# Imipenem/Cilastatin/Relebactam for Complicated Infections: A Real-World Evidence

**DOI:** 10.3390/life14050614

**Published:** 2024-05-10

**Authors:** Pasquale Sansone, Luca Gregorio Giaccari, Giusy Di Flumeri, Maria Caterina Pace, Vincenzo Pota, Francesco Coppolino, Simona Brunetti, Caterina Aurilio

**Affiliations:** 1Department of Woman, Child and General and Specialized Surgery, University of Campania “Luigi Vanvitelli”, 80134 Naples, Italy; pasquale.sansone@unicampania.it (P.S.); lucagregorio.giaccari@gmail.com (L.G.G.); mariacaterina.pace@unicampania.it (M.C.P.); vincenzo.pota@inwind.it (V.P.); francesco.coppolino1987@gmail.com (F.C.); simona.brunetti@studenti.unicampania.it (S.B.); 2UOC Emerging Infectious Disease with High Contagiousness, AORN Ospedali dei Colli P.O. C Cotugno, 80131 Naples, Italy; giusydiflumeri@ospedalideicolli.it

**Keywords:** imipenem/cilastatin/relebactam, recarbrio, complicated urinary tract infections, complicated intra-abdominal infections, hospital-acquired pneumonia, ventilator-associated pneumonia, multidrug-resistant

## Abstract

(1) Background: Infections caused by multidrug-resistant (MDR) bacteria represent one of the major global public health problems of the 21st century. Beta-lactam antibacterial agents are commonly used to treat infections due to Gram-negative pathogens. New β-lactam/β-lactamase inhibitor combinations are urgently needed. Combining relebactam (REL) with imipenem (IMI) and cilastatin (CS) can restore its activity against many imipenem-nonsusceptible Gram-negative pathogens. (2) Methods: we performed a systematic review of the studies reporting on the use of in vivo REAL/IPM/CS. (3) Results: A total of eight studies were included in this review. The primary diagnosis was as follows: complicated urinary tract infection (*n* = 234), complicated intra-abdominal infections (*n* = 220), hospital-acquired pneumonia (*n* = 276), and ventilator-associated pneumonia (*n* = 157). Patients with normal renal function received REL/IPM/CS (250 mg/500 mg/500 mg). The most frequently reported AEs occurring in patients treated with imipenem/cilastatin plus REL/IPM/CS were nausea (11.5%), diarrhea (9.8%), vomiting (9.8%), and infusion site disorders (4.0%). Treatment outcomes in these high-risk patients receiving REL/IPM/CS were generally favorable. A total of 70.6% of patients treated with REL/IPM/CS reported a favorable clinical response at follow-up. (4) Conclusions: this review indicates that REL/IPM/CS is active against important MDR Gram-negative organisms.

## 1. Introduction

Infections caused by multidrug-resistant (MDR) bacteria are some of the major global public health problems of the 21st century. These infections continue to increase and limit the utility of existing antibacterial agents. Bacterial antimicrobial resistance caused 1.27 million deaths worldwide in 2019 and contributed to 4.95 million deaths [1]. It is hypothesized that global antimicrobial resistance infections could cause 10 million deaths per year by 2050, surpassing cancer deaths [2].

MDR Gram-negative organisms (i.e., carbapenem-resistant Enterobacterales, carbapenem-resistant Pseudomonas aeruginosa, and extensively drug-resistant (XDR) Acinetobacter baumannii) are a particularly serious threat worldwide [3]. These organisms are important pathogens in complicated urinary tract infections (cUTIs), complicated intra-abdominal infections (cIAIs), and hospital-acquired pneumonia, including ventilator-associated pneumonia (HAP/VAP).

Beta-lactam antibacterial agents are commonly used to treat infections due to Gram-negative pathogens. Increasing resistance to beta-lactams, including the carbapenems, has led to some organisms being effectively untreatable or treatable only with recourse to colistin with or without other agents to which they remain at least partly susceptible. Increasing resistance to beta-lactams, including carbapenems, has led some organisms to be difficult to treat or cure only by using colistin in association with other agents to which they are at least partially sensitive [3].

The future management of antimicrobial resistance includes the development of new antimicrobials [4,5,6]. A common resistance mechanism in Gram-negative bacteria is the production of β-lactamases [7]. New combinations of β-lactam/β-lactamase inhibitor are urgently needed to treat multidrug-resistant pathogens [7,8].

Relebactam (REL) is a β-lactamase inhibitor of the diazabicyclooctane family [9]. REL is an inhibitor of class A (KPC- and extended-spectrum b-lactamase [ESBL]-producing) and class C β-lactamases, which are responsible for many carbapenem-resistant Pseudomonas aeruginosa and carbapenem-resistant Enterobacterales; it has no inhibitory activity against class B β-lactamases such as New Delhi metallo-blactamase (NDM) and imipenemase (IMP) [9].

When relebactam (REL) is associated with imipenem (IMI) and cilastatin (CS), its activity could be restored against many imipenem-nonsusceptible Gram-negative pathogens. IMI binds to penicillin binding proteins (PBPs), leading to the inhibition of cell wall peptidoglycan synthesis. CS limits the renal metabolism of IMI and has no antibacterial activity. REL is a non-β lactam inhibitor of Ambler class A and class C β-lactamases; it does not inhibit class B enzymes (metallo-beta-lactamases) or class D carbapenemases. REL has no antibacterial activity.

REAL/IPM/CS was approved for treating patients older than 18 years with infections caused by susceptible Gram-negative bacteria and with limited or no alternative treatment options: (1) complicated urinary tract infections (cUTIs); (2) complicated intra-abdominal infections (cIAIs); (3) hospital-acquired pneumonia (HAP), including ventilator-associated pneumonia (VAP); (4) bacteraemia as a complication of HAP or VAP; and (5) infections caused by bacteria classed as aerobic Gram-negative bacteria when other treatments might not work. REAL/IPM/CS was first approved by the U.S. Food and Drug Administration (FDA) on 16 July 2019 for cUTI and cIAI and then on 4 June 2020 [10,11]. It received authorization by the European Medicines Agency (EMA) on 12 December 2019 and then on 13 February 2020 [12,13].

Objectives. We performed a systematic review of the studies reporting on the use of in vivo REAL/IPM/CS. To the best of our knowledge, there are no previously published systematic reviews that have evaluated the use of in vivo REAL/IPM/CS in the treatment of MDR gram-negative organisms.

## 2. Materials and Methods

### 2.1. Protocol and Registration

This systematic review followed the PRISMA (Preferred Reporting Items for Systematic Reviews and Meta-Analyses) guidelines [14]. The protocol is available upon reasonable request. Registration in the International Prospective Register of Systematic Reviews (PROSPERO) was not carried out.

### 2.2. Literature Search Strategy

The main databases (PubMed, EMBASE, and Google Scholar) were screened to identify studies reporting on patients with complicated infections treated with REAL/IPM/CS. Other studies were identified from the reference lists. Terms used for the research were “*imipenem/cilastatin/relebactam*”, “*recarbrio*”, “*complicated urinary tract infections*”, “*complicated intra-abdominal infections*”, “*hospital-acquired pneumonia*”, “*ventilator-associated pneumonia*”, and “*bacteraemia*”. P.S. and L.G.G screened the titles and abstracts to identify the keywords and then the selected papers were read in full. If they disagreed, a third reviewer (M.C.P.) was consulted.

The initial search was conducted on 1 December 2023. All publications were included from inception through the end of November 2023.

### 2.3. Inclusion and Exclusion Criteria

Inclusion criteria were the following: (1) the full study was published; (2) the study described clinical use of REAL/IPM/CS for complicated infections; (3) the agent responsible for the infection was MDR Gram-negative organisms; and (4) the study reported the clinical outcome of the patient(s) treated with REAL/IPM/CS. Exclusion criteria were the following: (1) the study did not report clinical outcomes; (2) the study had duplicate data with others (in these cases, only the largest study was retained); and (3) the study presented pooled data that did not allow for extrapolation of useful information.

### 2.4. Data Extraction

Data analyzed were as follows:Patient demographics (sex, age, weight) and clinical characteristics (Acute Physiology and Chronic Health Evaluation [APACHE] II score). APACHE II is a severity-of-disease classification system, used to measure the severity of disease within 24 h of admission of an adult patient to an intensive care unit (ICU) [15].Clinical diagnosis, common baseline pathogens, and β-lactamases detected.Therapeutic regimen.Any drug-related AE, such as alterations of the blood and lymphatic system (eosinophilia, pancytopenia, neutropenia, leukopenia, thrombocytopenia, thrombocytosis), alterations of the nervous system (seizures, hallucinations, confusional states, myoclonic activity, dizziness, drowsiness), vascular alterations (thrombophlebitis, hypotension), gastrointestinal alterations (diarrhea, nausea, vomiting), hepatobiliary alterations (alanine aminotransferase increased, aspartate aminotransferase increased), alterations of the skin and subcutaneous tissue (rash, urticaria, itching), and alterations in diagnostic tests (increases in serum alkaline phosphatase, positive Coombs test, prolonged prothrombin time, decreased hemoglobin, increased serum bilirubin, increased blood urea nitrogen).Any drug-related serious AE. A serious adverse event (SAE) is defined as any adverse event that occurs at any dose and is life-threatening.Discontinuation due to a drug-related AE.Clinical response at the end of treatment (EOT) and at the follow-up. A favorable clinical response is based on the resolution of all or most of the signs and symptoms of the infection due to the therapy, or returning to the pre-infection state, and no other antibiotic therapy was performed.Microbiological response at the end of treatment (EOT) and at the follow-up. A favorable microbiological response is based on the eradication or presumptive eradication of all bacterial pathogens identified before the start of therapy.Overall deaths and drug-related deaths.

The risk of bias of each study included in this review was assessed using the Cochrane Collaboration’s bias assessment tool [16]. “High risk”, “low risk”, or “unclear” was assigned based on the following criteria: random sequence generation, allocation concealment, blinding of participants and personnel, blinding of outcome assessment, complete outcome data, elective reporting, and other biases.

### 2.5. Outcomes

The primary outcome was to evaluate the percentage of participants with a favorable clinical response at the end of REAL/IPM/CS administration and at the follow-up. The secondary outcomes were (1) number of participants with any drug-related AEs; (2) number of participants with any serious adverse event (SAE); (3) number of participants who discontinued REAL/IPM/CS due to a drug-related AE; (4) number of participants with a favorable microbiological response at the end of REAL/IPM/CS administration and at the follow-up; and (5) number of drug-related deaths.

A subanalysis of the RCTs was included in this review to analyze the efficacy of REAL/IPM/CS compared to different comparators.

### 2.6. Statistical Analysis

Continuous variables were extracted as medians and or means, depending on how they were presented in the original article. Clinical characteristics were reported together with the ratio of the number of patients in whom the variable was present (n) and the total number of reported cases (N): n/N (%). For the symptoms, we considered that they were absent rather than missing if they were not cited in the manuscript. Comparisons were performed using Student’s *t* test and the level of statistical significance was *p* < 0.05. The *p*-value is reported in this article only if statistically significant.

## 3. Results

A total of eight studies were included in this review [17,18,19,20,21,22,23,24]. The flow diagram shows the results from the literature search and the study selection process (see Figure 1).

### 3.1. Study Characteristics

In Table 1, all the studies included are reported in alphabetical order with a brief clinical description for each case. Except for two studies [18,22], all others were randomized controlled trials (RCTs). All studies were multicenter.

All studies were free from bias, except for one study due to its observational nature [20].

### 3.2. Patients

A total of 892 patients were included in this review. There were 703 males (60.8%) and 453 females (39.2%). The mean age was 59.3 ± 4.9 years (18–96 years). The mean weight was 74.0 ± 6.0 kg. The APACHE II score was reported for 334 patients and this was >15 in 147 (44%) patients.

Baseline characteristics of patients are reported in Table 2.

### 3.3. Type of Infection and Isolate

As shown in Table 3, the primary diagnosis was as follows: complicated urinary tract infection (*n* = 234), complicated intra-abdominal infections (*n* = 220), hospital-acquired pneumonia (*n* = 276), and ventilator-associated pneumonia (*n* = 157).

The main bacterial agents involved were *Acineto baumanii* (*n* = 34), *Bacteriodes* spp. (*n* = 40), *Enterobacter* spp. (*n* = 325), *Klebsiella pneumoniae* (*n* = 114), *Pseudomonas aeruginosa* (*n* = 136), *Proteus* spp. (*n* = 26), and *Streptococcus* spp. (*n* = 26).

Isolates with β-lactamases were reported only in three studies [17,18,21]. Detected β-lactamases included older spectrum β-lactamases (*n* = 34), ESBLs (*n* = 34), KPC (*n* = 13), AmpC (*n* = 57), and OXA-48 (*n* = 4).

Baseline concurrent bacteremia with any pathogen was reported for 37 patients.

### 3.4. Therapeutic Regimen

Patients with normal renal function received IV REL/IPM/CS (250 mg/500 mg/500 mg) over 30 min every 6 h for 5–14 days. For patients with renal insufficiency, dose adjustments were made based on estimated glomerular filtration rate (GFR).

### 3.5. Adverse events and Outcome

A total of 174 drug-related adverse events were reported. The most frequently reported adverse events were nausea (*n* = 20), diarrhea (*n* = 17), vomiting (*n* = 17), and infusion site disorders (*n* = 7). Serious adverse events occurred 13 times.

Twenty-four patients discontinued treatment due to REL/IPM/CS administration.

Clinical and microbiological outcomes are reported in Table 4. The clinical response was favorable in 486 patients at EOT and in 630 patients at follow-up. The microbiological response was favorable in 286 patients at EOT and in 445 patients at follow-up.

Death occurred in 121 patients, but only one episode was directly related to REAL/IPM/CS administration [19].

### 3.6. Subanalysis Comparing to Different Comparators

We conducted a subanalysis of the RCTs included in this review to analyze the efficacy of REAL/IPM/CS compared to different comparators (see Table 5). Two studies analyzed the efficacy of REAL/IPM/CS versus Imipenem/Cilastatin and Colistin [15,19]; two studies analyzed the efficacy of REAL/IPM/CS versus Imipenem/Cilastatin and placebo [18,21]; and two studies analyzed the efficacy of REAL/IPM/CS versus Piperacillin/Tazobactam [17,22].

When compared to Imipenem/Cilastatin and placebo, the proportions of subjects with a favorable clinical response was similar among the two treatment groups (80.7% in REAL/IPM/CS group versus 83.2% in IPM/CS group). Rates of drug-related AEs and discontinuation were similar in the two groups. The most commonly reported AEs were nausea, headache, and diarrhea. When fatal AEs occurred, they were considered not drug-related in both groups.

When compared to Imipenem/Cilastatin and Colistin, the proportions of subjects with a favorable clinical response was similar among the two treatment groups (72.9% in REAL/IPM/CS group versus 72.7% in IPM/CS + Colistin group). Drug-related AEs and discontinuations occurred similarly in the two groups. No death occurred due to study drugs.

When compared to PIP/TAZ, the proportions of subjects with a favorable clinical response was similar among the two treatment groups (64.3% versus 60%). The incidence of drug-related AEs was similar across both treatment arms. There were 58 AEs among patients receiving REAL/IPM/CS and 56 among those receiving PIP/TAZ. The most commonly reported AEs were nausea, headache, and diarrhea. There were a similar number of patients who discontinued due to a drug-related AE. A patient receiving REAL/IPM/CS died due to a drug-related AE, but no further information is reported by the authors [17].

The differences between the group treated with REAL/IPM/CS and the groups treated with the various comparators were not statistically significant for both clinical outcome and the incidence of adverse events.

## 4. Discussion

The study population of this review consisted of patients at increased risk of adverse treatment outcomes and death, as shown by 44% of enrolled participants with APACHE II scores >15. According to literature data, patients with an APACHE II score of 17 or higher admitted to the ICU are at high risk of mortality [25].

In this review, almost half of the patients had HAP; 17.6% of patients had VAP. The remaining half of patients were equally treated for cUTI (26.2%) and cIAI (24.7%). According to the World Health Organization reports [5], our data confirm that carbapenem-resistant Enterobacterales (36.4%), carbapenem-resistant Pseudomonas aeruginosa (15.2%), and carbapenem-resistant Klebsiella pneumoniae (12.8%) are among the most important MDR Gram-negative pathogens in complicated nosocomial infections. In 3.8% of cases, REL/IPM/CS was used in the treatment of Acineto baumannii infections; however, REL generally does not improve susceptibility to imipenem in Acinetobacter baumannii [26].

Multidrug resistance is a worldwide problem among Gram-negative bacteria and is usually associated with the production of β-lactamases [3]. The most important plasmid-encoded beta-lactamases include the older-spectrum β-lactamases such as TEM, the extended-spectrum beta-lactamases such as the CTX-M enzymes, and KPC β-lactamases.

According to prescribing information [27], REL/IPM/CS (250 mg/500 mg/500 mg) is administered intravenously over 30 min every 6 h for 5–14 days. For patients with renal insufficiency, dose adjustments are made based on the estimated glomerular filtration rate (GFR). REL/IPM/CS is a potential monotherapy agent.

The most frequently reported AEs occurring in patients treated with imipenem/cilastatin plus REL/IPM/CS were nausea (11.5%), diarrhea (9.8%), vomiting (9.8%), and infusion site disorders (4.0%). Other reported adverse reactions occurring in greater than or equal to 2% of patients treated with REL/IPM/CS were headache, increased alanine aminotransferase, increased aspartate aminotransferase, pyrexia, and hypertension [27]. Serious AEs occurred 13 times. The overall rate of discontinuation was low; only 2.7% of patients discontinued treatment due to REL/IPM/CS administration.

REL/IPM/CS was generally well tolerated, with 1.4% of patients presenting a serious drug-related AE, and few therapy discontinuations due to drug-related AEs.

Treatment outcomes in these high-risk patients receiving REL/IPM/CS were generally favorable. A total of 70.6% of patients treated with REL/IPM/CS reported a favorable clinical response at follow-up. A favorable clinical response at the end of treatment was achieved in 54.5% of patients treated with REL/IPM/CS. Eradication or presumptive eradication of all MDR Gram-negative organisms identified at baseline was reported in 32.1% of patients at the end of treatment and in 49.9% of patients at follow-up.

The overall mortality rate was 13.6% among patients included in this review, but only one drug-related death was reported.

According to our subanalysis, REL/IPM/CS is non-inferior to Piperacillin/Tazobactam, and Imipenem/Cilastatin alone or with Colistin for treating complicated infections in adults. These agents appeared well tolerated based on the incidence of AEs and discontinuation rate.

## 5. Limitations

A previous study highlighted the clinical response, microbiological response, and risk of adverse events related to REL/IMI/CS in the treatment of bacterial infections [28]. Our review adds references for its clinical application. However, this review has some limitations. First, it is based on a limited number of studies with few patients. Second, only three studies reported isolates with β-lactamases. Third, outcomes were not always consistent across studies.

## 6. Conclusions

Despite the development of new antibacterial agents, there is still an unmet need for antibacterial agents with an acceptable safety profile for treating adults with aerobic Gram-negative bacteria when other treatments might not work. Our review indicates that REL/IPM/CS is active against important MDR Gram-negative organisms. REL/IPM/CS seems, moreover, to have a safe profile. Therefore, REL/IPM/CS seems to be a useful alternative for the treatment of most infections due to MDR Gram-negative bacteria.

## Figures and Tables

**Figure 1 life-14-00614-f001:**
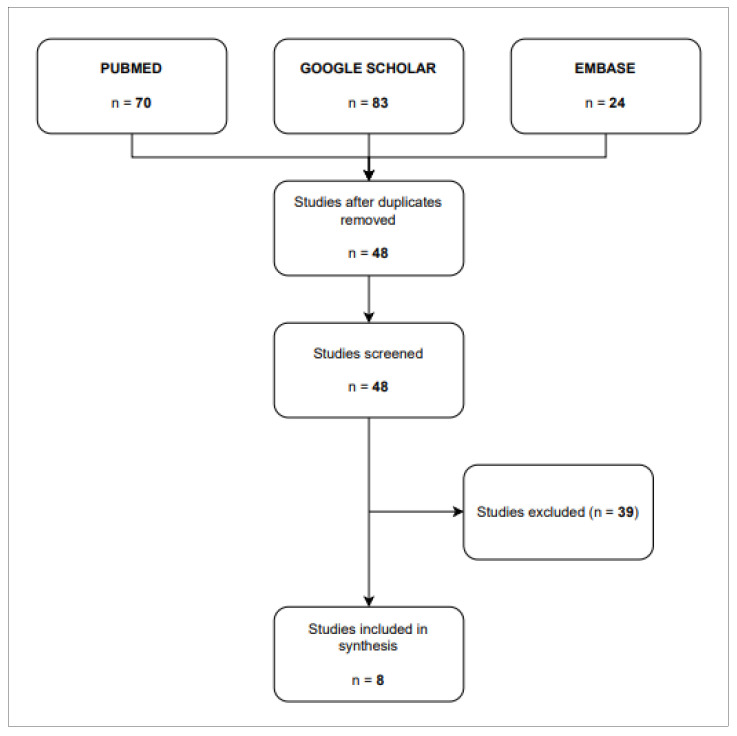
Flow diagram.

**Table 1 life-14-00614-t001:** Study characteristics.

Author, Year	Study Type	Comparator	No.	Country, Sites	Infection
Kayne, 2020 [15]	RCT	IMI + colistin	31	United States, multicenter	–cUTI *n* = 26 –cIAI *n* = 7 –HAP (VAP) *n* = 16 (14)
Kohno, 2020 [16]	CT	None	83	Japan, 29 sites	–cUTI *n* = 44 –cIAI *n* = 39
Li, 2023 [17]	RCT	PIP/TAZ	134	China, 8 sites	–HAP (VAP) *n* = 134 (45)
Lucasti, 2016 [18]	RCT	IMI + placebo	170	20 countries, 45 sites	–cIAI *n* = 170
Motsch, 2020 [19]	RCT	IMI + colistin	21	Brazil, 1 site; Columbia, 1 site; Estonia, 1 site; Germany, 1 site; Lithuania, 1 site; Mexico, 1 site; Peru, 1 site; Romania, 1 site; Turkey, 2 sites; Ukraine, 4 sites; United States, 2 sites.	–cUTI *n* = 11 –cIAI *n* = 2 –HAP (VAP) *n* = 8 (7)
Rebold, 2021 [20]	OS	None	21	United States, 8 sites	–cUTI *n* = 3 –cIAI *n* = 2 –HAP (VAP) *n* = 11 (?) –IPD/bone *n* = 4 –SSTI *n* = 1
Sims, 2017 [21]	RCT	IMI + placebo	150	Bulgaria, 4 sites; Greece, 2 sites; Latvia, 4 sites; Peru, 2 sites; Poland, 1 site; Romania, 5 sites; Russia, 2 sites; Republic of Korea, 2 sites; Turkey, 2 sites; Ukraine, 8 sites; United States, 2 sites.	–cUTI *n* = 150
Titov, 2021 [22]	RCT	PIP/TAZ	264	Argentina, 2 sites; Brazil, 5 sites; Bulgaria, 7 sites; Canada, 1 site; Colombia, 2 sites; Croatia, 1 site; Czech Republic, 1 site; Estonia, 2 sites; France, 5 sites; Georgia, 5 sites; Guatemala, 1 site; Italy, 2 sites; Japan, 26 sites; Latvia, 2 sites; Lithuania, 1 site; Mexico, 4 sites; Norway, 1 site; Philippines, 6 sites; Portugal, 3 sites; Romania, 3 sites; Russia, 8 sites; Serbia, 2 sites; Republic of Korea, 4 sites; Spain, 1 site; Turkey, 4 sites; Ukraine, 6 sites; United States, 8 sites.	–HAP (VAP) *n* = 264 (91)

cUTI, complicated urinary tract infection; cIAI, complicated intra-abdominal infection; CT, clinical trial; HAP, hospital-acquired pneumonia; IMI, imipem; OS, observational study; PIP/TAZ, piperacillin/tazobactam; RCT, randomized controlled trial; VAP, including ventilator-associated pneumonia.

**Table 2 life-14-00614-t002:** Baseline characteristics of patients.

No. of Patients	892
M/F	530/362
Age (years) ± SD	59.3 ± 4.9
Weight (kg) ± SD	74.0 ± 6.0
APACHE II	
>15	147 (44%)
≤15	187 (56%)

M, male; F, female.

**Table 3 life-14-00614-t003:** Primary diagnosis, baseline pathogens, and β-lactamases.

	No. (%)
Primary diagnosis	
–cUTI	234 (26.2)
–cIAI	220 (24.7)
–HAP	276 (31.0)
–VAP	157 (17.6)
–others	5 (0.6)
Baseline pathogens	
–*Acineto baumannii*	34 (3.8)
–*Bacteriodes* spp.	40 (4.5)
–*Enterobacter* spp.	325 (36.4)
–*Klebsiella pneumoniae*	114 (12.8)
–*Pseudomonas aeruginosa*	136 (15.2)
–*Proteus* spp.	26 (2.9)
–*Streptococcus* spp.	26 (2.9)
β-lactamases	
Class A	
Older spectrum β-lactamases	34
–SHV	8
–TEM	26
ESBLs	34
–SHV	4
–CTX-M	26
–VEB	0
KPC	13
Class C	
–PDC	51
–ACT	0
–CMY	3
–DHA	3
Class D	
–OXA-48	4

cUTI, complicated urinary tract infection; cIAI, complicated intra-abdominal infection; HAP, hospital-acquired pneumonia; VAP, including ventilator-associated pneumonia.

**Table 4 life-14-00614-t004:** Clinical and microbiological outcome.

	No. (%)
Clinical response	
–EOT	486 (54.5)
–Follow-up	630 (70.6)
Microbiological response	
–EOT	286 (32.1)
–Follow-up	445 (49.9)
Deaths	121 (13.6)
–Drug-related deaths	1

EOT, end of treatment.

**Table 5 life-14-00614-t005:** Subanalysis (comparator versus REAL/IPM/CS).

Study	Comparator (No.)	Favorable Clinical Response (No., %)	AEs (No.)
Kayne, 2020 [15] Motsch, 2020 [19]	IMI + colistin (33)	24 (72.7%)	5
Lucasti, 2016 [18] Sims, 2017 [21]	IMI + placebo (197)	164 (83.2%)	20
Li, 2023 [17] Titov, 2021 [22]	PIP/TAZ (403)	242 (60%)	56

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
