# Peer review of "Imipenem/Cilastatin/Relebactam for Complicated Infections: A Real-World Evidence"

_life, 2024, doi:10.3390/life14050614_

Round 1

Reviewer 1 Report (Previous Reviewer 1)

Comments and Suggestions for Authors

Dear Authors,

Thank you for submitting the revised version of your manuscript. I have also read your response to the different queries and suggestions. The new text and analysis have greatly improved the quality of the manuscript. However, there are still some pending aspects that should be addressed carefully.

Starting from the title, "A Real-World Evidence," this expression is quite common at present. Notwithstanding this, your study consists of a review of published evidence, and all evidence comes from RCTs or CTs, so experimental conditions. In my opinion, the text can't include any reference to RWE. This can be misleading.

I would suggest adding a new table to explain clearly the additional subanalysis included in the text.

Tables 2, 3 and 4 should be split according to the different comparators. Adding all studies is not informative.

An important reflection is that in different places of the manuscript, it says: "X is similar to Y", "AEs were similar to..." These kinds of expressions are not valid. Were these differences statistically significant or not? This is what the readers are waiting for. In the case of AEs, were there differences? If not, which statistical test has been used to say that?

One patient died in the group receiving the medicines of interest. It seems that this was the only death related to the medicines. Please, explain which was the reaction leading to death. Also, describe in detail the serious AEs.

Author Response

– "A Real-World Evidence," this expression is quite common at present. Notwithstanding this, your study consists of a review of published evidence, and all evidence comes from RCTs or CTs, so experimental conditions. In my opinion, the text can't include any reference to RWE. This can be misleading. → Modified in “Imipenem/Cilastatin/Relebactam for Complicated Infenctions: A Systematic Review.”
– table to explain clearly the additional subanalysis included in the text. → Added Table 5. Subanalysis (comparator versus REAL/IPM/CS).
Tables 2, 3 and 4 should be split according to the different comparators. Adding all studies is not informative.
– An important reflection is that in different places of the manuscript, it says: "X is similar to Y", "AEs were similar to..." These kinds of expressions are not valid. Were these differences statistically significant or not? This is what the readers are waiting for. In the case of AEs, were there differences? If not, which statistical test has been used to say that? → We added in Statistical Analysis “Comparisons were performed using Student's t test and the level of statistical significance was p<0.05. The p-value was reported in the article only if statistically significant.”; in Subanalysis “The differences between the group treated with REAL/IPM/CS and the groups treated with the various comparators were not statistically significant for both clinical outcome and the incidence of adverse events.”
– One patient died in the group receiving the medicines of interest. It seems that this was the only death related to the medicines. Please, explain which was the reaction leading to death. Also, describe in detail the serious AEs. → We added “A patient receiving REAL/IPM/CS  died  due to a drug-related AE, but no further information is reported by the authors [17].”

Reviewer 2 Report (Previous Reviewer 2)

Comments and Suggestions for Authors

After evaluating the modifications that the authors introduced in their manuscript, based on the review that I myself did previously, I consider that in this version, the manuscript meets the criteria to be published in this journal.

Author Response

Nothing to add.

Round 2

Reviewer 1 Report (Previous Reviewer 1)

Comments and Suggestions for Authors

Dear Authors,

Thank you for sending the new version of your manuscript and considering the suggestions done in the previous round. Now, the results are clearer.

kind regards

This manuscript is a resubmission of an earlier submission. The following is a list of the peer review reports and author responses from that submission.

Round 1

Reviewer 1 Report

Comments and Suggestions for Authors

Dear Authors,

Certainly, new antimicrobials (or combinations) with efficacy on AMR organisms are good news, and it is useful to review the available evidence to guide prescribers. So, I read your manuscript with interest.

Notwithstanding this, reading your manuscript raised many questions that remain unsolved in this manuscript. I will describe a few, but to my opinion, the whole manuscript should be carefully revised and rewritten.

1) Six out of eight studies identified are randomised clinical trials. So, it is necessary to describe the different comparators, the number of patients included in each arm, and the outcomes and adverse events detected in each study. Adding the patients exposed to the medicine of interest and analyzing them as a homogeneous group is inappropriate.

2) One of the relevant points is adverse reactions reported in the patients who have received the combination of interest. Again, to be able to have an insight about the safety of the combination, we must know what happened in each study with the patients receiving the comparator. So, for example, in the study published by Li and cols., Table 3, the proportion of patients with serious AE in the study group was 21%, while it was only 15% in the comparator group.

In your manuscript, only one sentence (repeated in the abstract) says that "Serious AEs occurred 13 times". In another paragraph, this number is qualified as "few". But, according to the figures given, 13 patients out of the 892  included in your analysis, this means 1.4% of patients in the sample. In my opinion, 1.4% of patients presenting a serious adverse event is something to be analysed carefully.

3) In different parts of the manuscript, you refer to the patients who died. The question here ism how many patients died in the comparator groups? I understand that these pathologies are serious and, in many cases, fatal. But, concerning the study you are proposing, the key question is: what happened with the patients of these studies that you are not showing because they were allocated to the comparator arms?

4) In addition to the previous comments, which require revising the whole text and tables, it is important to show data clearly in the tables and the text. The first time that a number is included between parentheses and the formula is repeated for different characteristics in the same sentence; it should be clarified what you are talking about in the first parentheses. For example: "A total of 8 studies were included in this review. Primary diagnosis was: 18 complicated urinary tract infection (n = 234), complicated intra-abdominal infections (n = 220), hos-19 pital-acquired pneumonia (n = 276), and ventilator-associated pneumonia (n = 157)." The text must say: "A total of 8 studies were included in this review. Primary diagnosis was: 18 complicated urinary tract infection (n = 234 patients), complicated intra-abdominal infections (n = 220), etc.".

This mistake is repeated in different variables along the text.

5) Tables must be self-explanatory. So, each time you include an abbreviation in a table, the table must have a footnote describing each abbreviation, independently of the fact that the abbreviation has already been explained in the text. Tables and text are different aspects of communication because many tables are taken apart from the text.

Of course, the discussion and the abstract will be different with the new way of presentation of the results.

Reviewer 2 Report

Comments and Suggestions for Authors

Thank you very much for allowing me to review this article.

From a methodological point of view, the article is well structured and executed, however it seems to me that from the point of view of analysis, this work has limitations.

On the one hand, a limitation to which the authors refer is the low number of studies included and patients analyzed.

On the other hand, 7 of the 8 studies included in the review have a clinical trial design. In these primary studies, relative risks and risk reductions of the study treatment compared to the comparator are analyzed in terms of efficacy and adverse events. However, nothing of this appears in the results of this work. It would be interesting to carry out an analytical study and not just a descriptive one of the primary studies. This would give greater validity to your results.

There is also no analysis of heterogeneity of the included studies.

And finally, in the discussion section, there is no discussion of the method, so it would be essential to include it.